# Plasma-Treated Nitrogen-Enriched Manure Does Not Impose Adverse Effects on Soil Fauna Feeding Activity or Springtails and Earthworms Abundance

Hesam Mousavi [1,*], Thomas Cottis [1], Reidun Pommeresche [2], Peter Dörsch [3] and Svein Øivind Solberg [1]

[1] Faculty of Applied Ecology, Agricultural Sciences, and Biotechnology, Inland Norway University of Applied Sciences, 2418 Elverum, Norway

[2] Norwegian Centre for Organic Agriculture, 6630 Tingvoll, Norway

[3] Faculty of Environmental Sciences and Natural Resource Management, Norwegian University of Life Sciences, 1432 Aas, Norway

[*] Correspondence: hemousavi65@gmail.com

**Abstract:** Plasma treatment of animal manure is a new technology, enriching the manure with plant-available nitrogen. Therefore, the product is termed nitrogen-enriched organic fertilizer (NEO). The producer (N2 Applied) claims that NEO can be a sustainable alternative to conventional fertilizers used in agriculture. However, the effect of this product on soil-dwelling organisms is unknown. This study investigates and compares the effects of NEO on changes in soil fauna feeding activity, the abundance of springtails, and the abundance and weight of earthworms to mineral fertilizer, organic fertilizer (cattle slurry), and no fertilizer in pot and field experiments with sandy clay loam soil. Early effect evaluation (week 7) indicated influences on soil fauna feeding activity; among treatments, higher amounts of fertilizers went along with lower feeding activity, regardless of fertilizer type. However, the initial fertilizer application stimulation was transient and stabilized with time after fertilization towards mid-term (week 14) and late effect evaluations (week 21). Accordingly, differences between feeding activities were less than five percent at late effect evaluation. Similarly, none of the fertilizers used imposed adverse effects on the abundance of springtails and the abundance and weight of earthworms; these parameters were almost identical among all fertilizing treatments. After two years of application in field trials and in a pot experiment, NEO and the other used fertilizers seem not to harm the selected soil-dwelling organisms.

**Keywords:** sustainable agriculture; nitrogen; fertilizing; organic farming; soil fauna; NEO

## 1. Introduction

The current trends of population growth and resource scarcity underline the importance of using new sustainable technologies in agriculture. Agri-food systems worldwide depend severely on mineral fertilizers [1], and current plant production systems are intensively fertilized with nitrogen (N) [2,3]. According to the FAO, the global nitrogen input into agriculture is eight times higher today than in the 1960s [4].

The favorable effect of N fertilization on plant productivity is well recognized [5–7]. However, N intensification has many severe trade-offs [3,8]. Although increased food production is crucial for sustaining an increasing human population, preserving soil fertility is also critical. While increasing food production per area is commonly highlighted, the effects of excessive fertilizing on soil organisms and their functions are often neglected [9,10]. Overuse of fertilizers can lead to air, soil, and water pollution, as well as adverse effects on biodiversity and the climate [11]. Besides, soil nutrients are manipulated through fertilization, and changes in functional soil groups are stimulated by favoring some groups over others [12]. Soil fauna and soil microorganisms contribute to various ecosystem services such as plant health, disease protection, pathogenicity, and nutrient turnover [13].

Considering the European Green Deal and the "Farm to Fork strategy," the European Commission aims for at least a 50% reduction in nutrient leaching by 2030. Accordingly, a 20% reduction in fertilizer use is anticipated [11]. Moreover, identifying fertilization regimes with the least possible adverse impact on soil organisms is fundamental as it enhances sustainability in food production. Hence, there is a necessity for the purposeful use of fertilizers, e.g., fertilizing with mineral and organic fertilizers, fertilizer efficiency enhancement [14], and developing novel high-tech fertilizers.

Nitrogen Enriched Organic fertilizer (NEO) has been introduced as a novel fertilizer with potentially advantageous properties [15]. Atmospheric nitrogen is fixed as nitrogen oxides (NOx) by a plasma process using green electricity and added to organic fertilizers (e.g., manure, slurry, digestate) using N2 Applied's (Asker, Norway) patented unit. Once the NOx reacts with water, it forms nitrous acid ($HNO_2$) and nitric acid ($HNO_3$), which lowers the pH of the slurries and stabilizes it. The units are small enough to allow farmers to produce their NEO locally, resulting in self-sufficiency and enhanced agricultural sustainability when substituting conventional organic and mineral fertilizers, nonetheless because of lowered ammonia ($NH_3$) and methane ($CH_4$) emissions [15–17]. NEO is highly fluid, holding less than 10% solid particles due to filtration during the production process. In the present study, we investigate and compare the effects of NEO on soil living organisms with those of other fertilizers.

Hypothetically, mineral fertilizers can enhance soil biological activity by increasing plant productivity and residue return [10,18]. Studies on fertilizer effects on the abundance and weight of earthworms found that a combination of mineral and organic fertilizers was even more significant than mineral fertilizer alone [19–22]. Another study indicated a positive effect of mineral fertilizer on springtails and mite abundance, despite the reduction in species richness [23]. On the contrary, N fertilizers, mainly ammonium N, can potentially contribute to diminishing soil biological activity by acidifying soil and inducing changes in soil functional communities [10,18,24,25]. Besides, repressing certain soil enzymes involved in nutrient cycles, e.g., the amidase involved in the N cycle, is likely due to the repeated application of mineral fertilizers [24].

Similarly, perhaps due to reduced plant species richness, reduced soil microbial weight was reported in perennial grassland under high N fertilizing rates. However, results from annual croplands do not support this conclusion [9,26–28], despite results being highly dependent on fertilization rates [18,21]. Regardless, the functional activity of soil organisms is a complex trait controlled by a multitude of environmental and management factors and recurrent mineral N fertilization [25]. It was shown that repeated application of organic fertilizers stimulates soil microbial and faunal growth and activity [18,19,29,30]. Indeed, organic amendments provide carbon for soil living communities and improve productivity and residue return. In addition, organic fertilizers enhance soil microbial and faunal communities more than chemical fertilizers. Albeit, this positive effect is expressed more when combining organic and mineral fertilizers [19,20,29]. However, these effects vary between annual and perennial production systems [29].

The susceptibility of soil fauna and invertebrates to elevated nitrogen levels differs. For example, soil fauna feeding activity under fertilization has been reported to be reduced in the short-term [9,24] and the long term [31]. Another study found that the abundance of springtails decreased following cattle slurry application; however, it recovered, but not entirely to initial numbers, later during the same growing season [32]. Furthermore, there are positive reports about fertilizers enhancing soil faunal structure, diversity, and feeding activity [26,27,31], specifically for springtails in the topsoil layer [33]. The mentioned controversies highlight the importance of investigating fertilizer effects on soil biota, especially when dealing with novel fertilizers such as NEO.

The current study aimed to (1) identify if NEO has any detrimental effects on soil fauna compared to conventional fertilizers and (2) develop a method for evaluating the immediate effect of fertilizers on earthworm abundance and weight. A preliminary study showed that NEO did not negatively affect soil fauna feeding activity more than other

fertilizers [34]. In the present study, we expand our research using a different type of NEO and compare effects on the abundance of earthworms and springtails as "bioindicators of soil quality" [35–37]. Fertilization treatments included mineral fertilizer, NEO, untreated cattle slurry, and a combination of organic and mineral fertilizers.

## 2. Materials and Methods

In this study, we conducted three sets of experiments. First, a growing chamber experiment to identify and compare the fertilizer effects on soil fauna feeding activity; second, a field experiment to identify and compare fertilizer effects on the abundance of springtails. Third, an outdoor experiment to identify and compare the immediate effect of fertilization on the abundance and weight of earthworms.

### 2.1. Soil Fauna Feeding Activity
2.1.1. Experimental Design

We conducted pot experiments in a growing chamber at Inland Norway University of Applied Sciences. Perforated pots ($13 \times 18$ cm$^2$ with 2.5 L of field soil were used. Treatments were distributed randomly among the four replicates after the pots were fertilized at loading time.

The trial consisted of five fertilization regimes distributed in seven fertilization treatments (Table 1); no fertilizer; mineral fertilizer (Yara Mila 18-3-15) [38]; NEO type D (N2 Applied) [15]; organic fertilizer (untreated cattle slurry); and organic fertilizer + mineral fertilizer (Yara Liva 16-0-0) [39]. NEO and untreated slurry were applied in liquid form, while Yara Mila and Yara Liva were pelleted. Treatments were (1) no fertilizer, (2) mineral fertilizer 73 kg N ha$^{-1}$, (3) mineral fertilizer 175 kg N ha$^{-1}$, (4) NEO type D 73 kg N ha$^{-1}$, (5) NEO type D 175 kg N ha$^{-1}$, (6) organic fertilizer 73 kg N ha$^{-1}$, (7) organic fertilizer + MF 175 kg N ha$^{-1}$.

**Table 1.** Fertilizing treatments and application rates used in the growing chamber trial.

| | Fertilizing Treatment | Organic Fertilizer (Tons ha$^{-1}$) | Kg N in Yara Mila18-3-15 (kg ha$^{-1}$) | Kg N in Organic Fertilizer (kg ha$^{-1}$) | Kg N in Yara Liva 16-0-0 (kg ha$^{-1}$) | Total kg N (kg ha$^{-1}$) |
|---|---|---|---|---|---|---|
| 1 | No fertilizer | - | - | - | - | 0 |
| 2 | Mineral fertilizer 73 kg N ha$^{-1}$ | | 73 | | | 73 |
| 3 | Mineral fertilizer 175 kg N ha$^{-1}$ | | 175 | | | 175 |
| 4 | NEO type D 73 kg N ha$^{-1}$ | 22 | | 73 | | 73 |
| 5 | NEO type D 175 kg N ha$^{-1}$ | 50 | | 175 | | 175 |
| 6 | Organic fertilizer 73 kg N ha$^{-1}$ | 55 | | 73 | | 73 |
| 7 | Organic fertilizer + mineral fertilizer 175 kg N ha$^{-1}$ | 55 | | 73 | 102 | 175 |

NEO type D had a pH of 5.22 and contained 1746 mg L$^{-1}$ NH$_4^+$ –N, 1131 mg L$^{-1}$ NO$_2^-$ –N, and 1562 mg L$^{-1}$ NO$_3^-$ –N, totaling 4439 mg L$^{-1}$ N. The untreated slurry had a pH of 7.32, containing 1804 mg L$^{-1}$ NH$_4^+$ –N and 149 mg L$^{-1}$ NO$_2^-$ –N, totaling 1953 mg L$^{-1}$ N. Therefore, we targeted a slurry amount of 55 tons ha$^{-1}$; nonetheless, during production, the N2 applied apparatus excludes all dry materials bigger than 5 mm; as a result, NEO's quantity decreases by 10% to 50 tons ha$^{-1}$. Therefore, each ton of untreated slurry contained 1.95 kg of plant-available N, while each ton of NEO contained 4.44 kg of plant-available N.

The soil was acquired from the adjacent experimental farm and analyzed at Eurofins soil lab (https://www.eurofins.no/agro-testing/ (accessed on 20 July 2022)), indicating a sandy clay loam texture, more than 10% clay, and soil organic matter of 4.5%. The soil pH was 7.4, which is relatively high, with a normal phosphorus status (P-AL = 11 mg/100 g), and a low potassium status (K-AL = 5 mg/100 g). Moreover, we estimated the soil's field capacity at 33.6% VWC, with a total pore volume of 41.4%.

In order to simulate field conditions, we planted seeds in the pots. Pots were prepared following the protocol that we developed before [34]. First, a soaked paper tissue was laid at the lowermost of the perforated pots to prevent soil outpour. Then, an initial 0.6 L (5 cm) soil load into the pots. Next, a soil load of 0.8 L (6 cm) was mixed with the fertilizer. Fertilizers were dosed following the advised field application rates (measured in tons per hectare), accounting for the soil surface in each pot (169 cm$^2$) (Table 1). Afterward, three rows of Italian ryegrass seeds (*Lolium multiflorum Lam.*), variety "Barpluto" (NAK Nederland/Ref. DE148-214011) per pot were then sown over the top of 0.9 L (6 cm) of additional soil. Finally, 0.2 L (1 cm) of soil was added to form the surface soil.

In the growing chamber, we used Lumatek ATS300W 80 × 80 cm LED light pads (https://lumatek-lighting.com/ (accessed on 21 July 2022)) that delivered a complete visible light spectrum (380–780 nm wavelength) recommended for plantation under controlled conditions [40]. Three adjacent LED pads were positioned 35 cm above the plants and were uplifted alongside plant growth. The light and dark intervals were adjusted according to Nordic summer days with 16 h light and 8 h darkness. Additionally, 16 pots per LED pad were confirmed to receive equal light using a digital light intensity meter. Throughout the experiment, the growth chamber had a temperature of 16 °C.

Five hundred milliliters of water, or 55% of the field capacity for our dry soil, was used to irrigate the pots at first since the soil was not entirely dry at pot preparation. After that, pots were irrigated with 200 mL of water thrice a week for the first four weeks. However, as plants progressed in the developmental stages, irrigation frequency was increased the weeks before harvest upon visual inspection [41].

The pots were positioned for the first two weeks adhering to each other. However, from week three, there was a five cm distance between the pots to avoid plants competing for light and space. Thinning was performed following germination; 24 vigorous plants per pot were kept (3 × 8 rows). Moreover, a few germinated weeds were removed by hand.

### 2.1.2. Evaluating Feeding Activity

Bait-lamina strips (Terra Protecta GmbH, Berlin, Germany) [42] were used to evaluate soil fauna feeding activity. This method is considered efficient, rapid, and reproducible with high statistical applicability [43,44]. This method evaluates the functional activity of soil fauna, feeding activity as one of the critical factors in soil nutrient cycling [44–46]. The technique has helped researchers screen various soil management practices and has given valuable information on the feeding activity of soil fauna. [44]. In this method, 16 1.5 mm diameter holes are located 5 mm apart on perforated PVC strips (1 mm × 6 mm × 120 mm). The holes are filled with bait substrate. The bait substrate comprises 5% activated carbon, 25% wheat bran, and 70% cellulose powder [47]. After a certain period of exposure to soil, the degree to which substrate is used up in the holes reveals the feeding activity of soil fauna, whereas soil microorganisms (e.g., bacteria, nematodes, fungi) have a negligible effect [42,48–50].

We conducted preliminary tests to determine proper intervals for the bait-lamina sampling in a pot experiment. In these tests, we noticed that after four weeks of soil fauna feeding activity, the percentage of bait consumption in the strips varied from 3–29%. Moreover, after extending the period to eight weeks, we had several strips with all holes empty, showing 100% feeding activity. Therefore, we determined seven weeks as an appropriate test period.

In this experiment, we planted three Bait-lamina strips diametrically in each pot (replicate) when watering for the first time to assess and compare the early effect of fertilization

on soil fauna feeding activities [47]. Seven weeks later, the first set of strips was taken out, and the second set was inserted in the same order as the first set to evaluate the mid-term effect. Seven weeks later (week 14 after plantation/fertilizing), the second set was taken out, and the third set was inserted to evaluate the late effect of fertilizing soil fauna feeding activity. At last, this set was taken out seven weeks later (week 21 after plantation/fertilizing). Plant growth and watering were sustained until the experiment's termination to simulate the conditions seen in an actual field. The strips were visually examined for the removal of the bait substrate on each sampling [47]. Three categories—empty (1), partially empty (0.5), or filled (0)—were assessed and used to describe the disappearance of the bait substrate [9]. With a maximum of 100% feeding activity (all 16 empty holes), each empty hole (score 1) was equivalent to 6.25% feeding activity.

### 2.2. The Abundance of Springtails (Collembola)

Springtails live in all soil layers, depending on soil moisture, and have diverse life forms in different soil strata and nutrition types [51]. They graze on fungi, algae, and bacteria or feed on plant detritus or other organic substances. [52]. They are great soil bio-indicators, especially in shallow soils. As a pray for other arthropods, they play a central role in the food chain [35].

### 2.2.1. Experimental Design

We conducted springtail sampling in two different field trials that had been fertilized with NEO and other fertilizers one year before the first sampling. The first trial was a cereal field located at Blæstad experimental farm at Inland Norway University of Applied Sciences (60°49′11.7″ N 11°10′48.4″ E). The second trial was on a grass field located at Stjørdal, Trøndelag (63°20′33.4″ N 10°17′56.9″ E). The experimental design in both trials was a traditional randomized complete block design with four replicates. Both trials consisted of four fertilization regimes; mineral fertilizer (Yara Mila 18-3-15) [38]; NEO type B (N2 Applied) [15]; organic fertilizer (untreated cattle slurry); and no fertilizer. Both fields were fertilized for two consecutive years. The grain field was fertilized once a year: before sowing: (22 April 2020 and 27 April 2021). The grass field was fertilized twice a year: in early spring (27 April 2020 and 4 May 2021) and after the first harvest (24 June 2020 and 15 June 2021).

Fertilizer doses in the grain field were (1) mineral fertilizer 666.6 kg ha$^{-1}$, (2) NEO 37.6 tons ha$^{-1}$, (3) organic fertilizer 41 tons ha$^{-1}$, and (4) no fertilizer. Then again, doses in the grass field were (1) mineral fertilizer 650 kg ha$^{-1}$ in spring + 500 kg ha$^{-1}$ after the first harvest, (2) NEO 37.5 tons ha$^{-1}$ + 28 tons ha$^{-1}$ after the first harvest, (3) organic fertilizer 41 tons ha$^{-1}$ + 30.5 tons ha$^{-1}$ after the first harvest, and (4) no fertilizer. NEO type B had a pH of 5.35 and contained 1480 mg L$^{-1}$ NH$_4^+$ –N, 777 mg L$^{-1}$ NO$_2^-$ –N, and 1250 mg L$^{-1}$ NO$_3^-$ –N, totaling 3507 mg L$^{-1}$ N. The cattle slurry used in this experiment had a pH of 7.32, and it contained 1804 mg L$^{-1}$ NH$_4^+$ –N and 149 mg L$^{-1}$ NO$_2^-$ –N, totaling 1953 mg L$^{-1}$ N.

NEO and untreated slurry were applied in liquid form while mineral fertilizer was pelleted. For a homogenous liquid fertilizer, all the barrels were stirred well prior to bottling/spreading to dissolve the sediments. Next, the fertilizers were dispersed manually using containers and rapidly harrowed with the soil using a tractor before sowing in the grain field and spreading on the grass field surface. The grain field was seeded with barley 'Rødhette' (180 kg ha$^{-1}$) in 2020 and spring wheat 'Mirakel' (220 kg ha$^{-1}$) in 2021. The grass field was a mixture of timothy, meadow fescue, and red clover seeded in 2019. In the grain field, herbicides were applied once in June with Ariane S (Corteva Agriscience, Puerto Rico) and once at the end of the growing season with Roundup (Bayer, Germany). No irrigation was applied. The 2021 season at Blæstad was decent regarding cereal growth, with a relatively cool May and a little over average precipitation: 78 mm in May and 62 mm in June. Moreover, in Trøndelag, the season was good, with precipitation around normal.

The soil in the field trial at Blæstad was identical to the soil we used in the growing chamber experiment (see Section 2.2.1). The soil in the grass field trial at Trøndelag was classified as clay loam. The organic matter content was 5.1%, pH was 6,2, and plant available phosphorus and potassium were normal (P-AL = 8, K-AL = 7), but the potassium reserve was high ($KHNO_3$ = 140).

### 2.2.2. Evaluating the Abundance of Springtails

The soils from experimental plots in the grain and grass fields were sampled twice in 2021 for springtail abundance. The first sampling occurred on 15 June 2021, some weeks after fertilization. The second sampling was on 20 October 2021, after harvesting. The temperature on the first sampling day in June was 20 °C and 6 °C after some rainy days in October when the soil was sampled again.

Three diametric samples were collected on each replicate's corners and in the center of the field plots. First, soil sampling was conducted by hammering down a corer (5 cm high, 5 cm diameter = 98.17 $cm^3$ volume) in the surface soil and collecting the sample into a zipper bag using a spade [35]. The samples were transferred directly to the lab, and the three samples from each replicate were placed upside down on slightly modified Berlese funnels [35,53,54]. In this method, moisture, heat, and light gradient drive the soil organisms to move away from the heat source (60 W lamp) [55], passing a mesh screen and falling into the vessel, ending in the collection tube filled with 91% ethanol. This way, animals can be preserved for further investigation for a long time. The samples remained in extraction units for a week on each sampling occasion. Following completion of extraction, the abundance of springtails in samples was counted and registered using a light microscope. Springtail abundances were scaled up to 1 $m^2$ and −5 cm depth, estimating the number of soil cores fitting into 1 $m^2$ (169.8).

### 2.3. The Fate of Earthworms (Lumbricidae)

Earthworms are medium to large oligochaetes that are substrate feeders and play an essential role in decomposition in the soil [35]. Due to several essential functions, their activity increases soil fertility. However, their distribution highly depends on moisture, soil type, pH, and vegetation. Earthworms are categorized into three ecological groups: litter dwellers, horizontal burrowers, and deep burrowers [56]. Because of their size, a significant fraction of the biomass in loamy meadows is composed of earthworms; however, they are scarce in shallow or acid soils [57,58]. In this study, we developed a protocol for evaluating the immediate effect of fertilizers on earthworms. The earthworms used in the experiments were a mixture of juveniles and adults from the most common Norwegian earthworms: geophagous (soil eating) field worms (*Aporrectodea caliginosa*) and pink worm (*Aporrectodea rosea*). In addition, other common species in Norwegian arable soil include dew worm *Lumbricus terrestris*, *L. rubellus*, and a few individuals of the less common *Allolobophora chlorotica* [30]. The earthworms used in our trials were found in an organic vegetable garden adjacent to the experimental field.

### 2.3.1. Experimental Design

The experiment was repeated twice in June 2021 and June 2022, with three replicates. The study location was at Blæstad experimental farm, Innlandet (60°49′11.7″ N 11°10′48.4″ E); the soil analysis was identical to the growing chamber experiment (Section 2.1.1), and the fertilizer treatments were (1) no fertilizer, (2) mineral fertilizer (Yara Mila) 666 kg ha$^{-1}$, (3) NEO type B (2021), and type D (2022) 3.4 tons ha$^{-1}$, and (4) organic fertilizer (untreated slurry) 3.7 tons ha$^{-1}$. Over, NEO and mineral fertilizer contained almost equal N per hectare in both experiments. The duration of the experiments was eight days.

### 2.3.2. Changes in Abundance and Weight of Earthworms

The developed protocol is as follows:

1. Holes with 30 cm diameter and 20 cm are dug out in the field. The soil from the holes was visually inspected to exclude present earthworms.
2. Earthworm-proof but water-permeable textile (tested before the experiment) is inserted into the hole.
3. Earthworms (Lumbricidae) used in the experiment were excavated from the same experimental farm two days before and stored in a pile of soil pending the experiment. On the day of starting the experiment, the earthworms were detached from the soil pile, sorted, weighed, and an equal number of worms (11 in the 2021 trial and 13 in the 2022 trial) making up a similar total weight were deposited in separate containers and marked. The worms were not rinsed before weighing. The worms were handled cautiously and remained detached from the soil for the shortest possible period. After counting and weighing, earthworms were transferred to other containers with soil.
4. Next, 10 cm of soil was filled back into the holes, and the earthworms were placed over the top.
5. The next 5 cm of soil was carefully mixed with the fertilizer and filled back into the hole.
6. As a supplementary food source for the worms, 100 g of grass was spread on this layer.
7. Two cm loose soil scattered over.
8. Finally, the last 3 cm of loose soil was scattered on the top.
9. Outer edges of the textile were fetched together and closed over. At last, a heavy substance (a stone) was placed over the top to inhibit wind opening or bird feeding. Finally, the experimental units were covered with white plastic tarpaulin on days of intense sun to avoid excessive temperature caused by the sun and the black textile.
10. At the end of the experiment (8 days), the soil bags were lifted out of the soil and dispersed over a flat surface. The living earthworms were carefully handpicked, counted, and weighed in less than five minutes to avoid desiccation.

### 2.4. Data Handling and Statistical Analyzes

We registered and sorted the data from each respective experiment. Using Minitab 20 statistical software (2021 Minitab, LLC (State College, PA, USA)), the differences in soil fauna feeding activity, the abundance of springtails, and variations in weight and abundance of earthworms were examined. The differences among fertilizing treatments were assessed using a one-way ANOVA and Welch's test. Games–Howell pairwise comparison was further utilized to compare, categorize, and plot data at a 95% confidence interval for the means. The error bars in the graphs are calculated by using individual standard deviations.

### 3. Results

### 3.1. Soil Fauna Feeding Activity

Using a growth chamber experiment where all variables except fertilization were held constant, we examined and evaluated the impact of different fertilization treatments on the feeding activity of soil fauna. We investigated the early effects (seven weeks), mid-term effects (14 weeks), and late effects (21 weeks).

There was a significant early fertilization effect on the feeding activity ($p = 0.001$). However, this effect was not associated with the type of fertilizer but with the amount of fertilizer applied (Figure 1A, Table S1). After seven weeks, mineral fertilizer 73 kg N ha$^{-1}$ (78.13%), organic fertilizer 73 kg N ha$^{-1}$ (74.74%), NEO type D 73 kg N ha$^{-1}$ (73.70%), and organic + mineral fertilizer 175 kg N ha$^{-1}$ (61.98%) exhibited increased soil fauna feeding activity relative to no fertilizer (54.17%). Both NEO type D 175 kg N ha$^{-1}$ (49.22%) and mineral fertilizer 175 kg N ha$^{-1}$ (46.35%) tended to have lower soil fauna feeding activity than soil without fertilizer. Additionally, the lead was insignificant even though the mixture of organic and mineral fertilizers was the only high N content treatment that improved soil fauna feeding activity above that of no fertilizer (Figure 1A, Table S1).

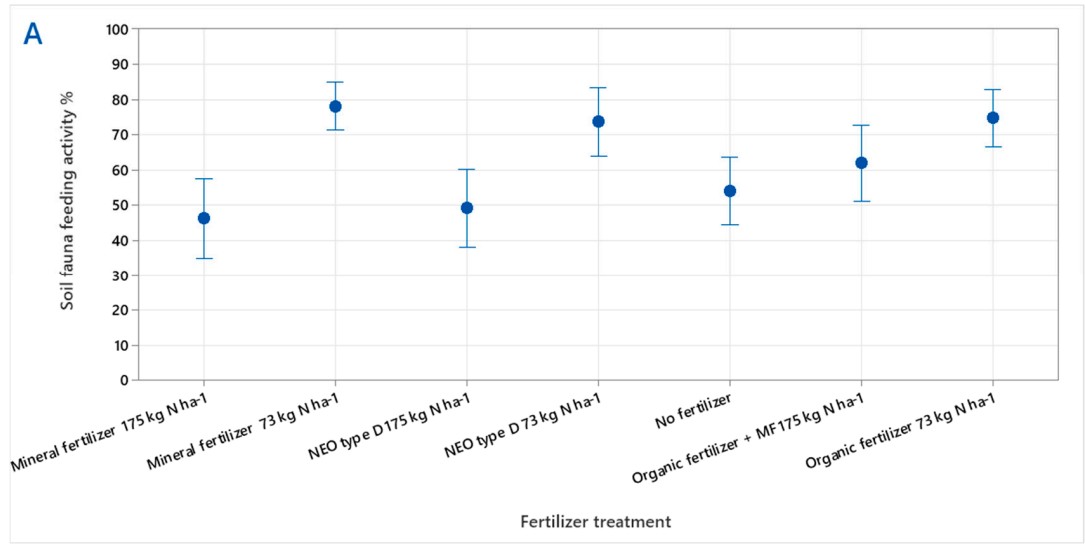

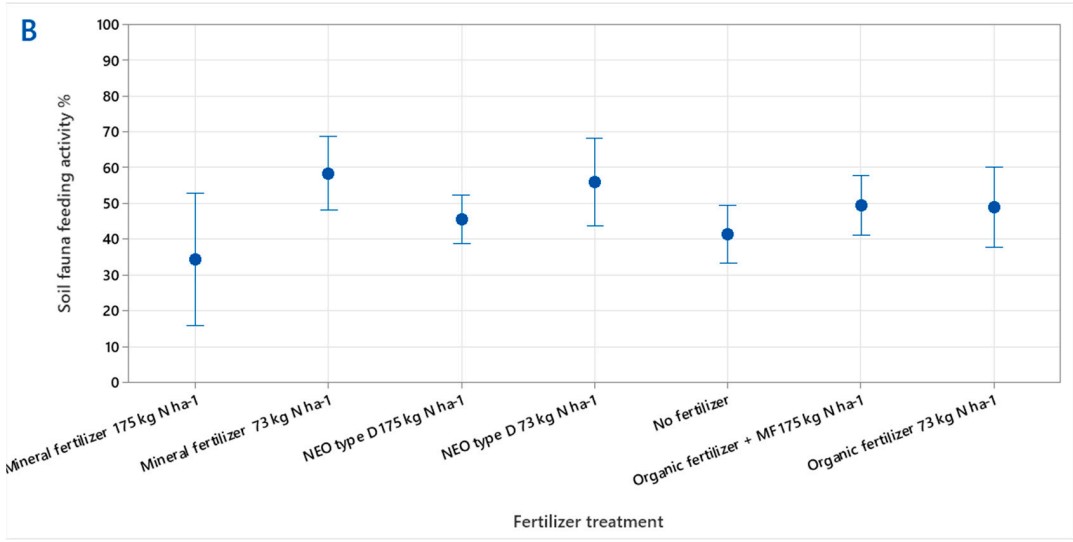

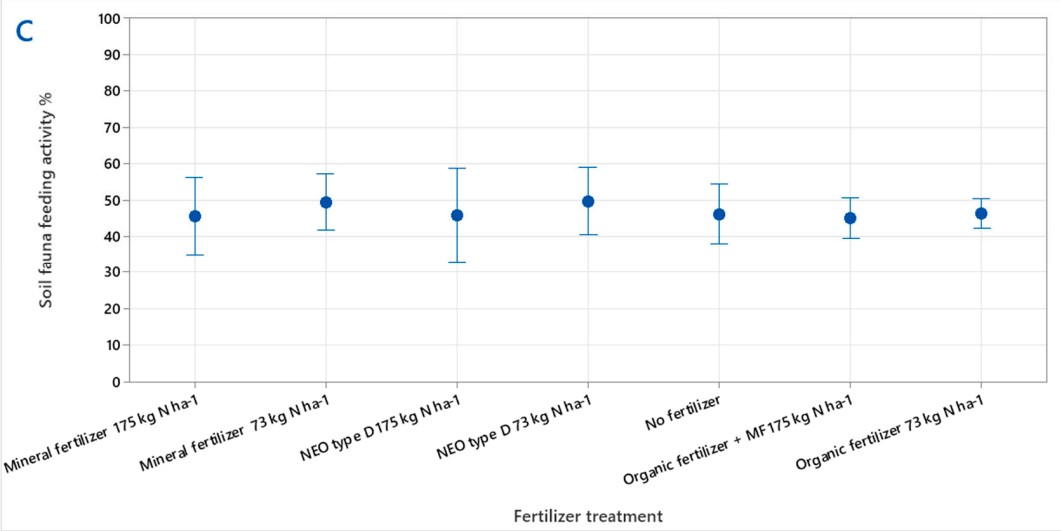

**Figure 1.** Effects of different fertilization treatments on soil fauna feeding activity (%) (**A**) seven weeks after fertilizing, (**B**) 14 weeks after fertilizing (7–14 weeks), and (**C**) 21 weeks after fertilizing (14–21 weeks). Error bars are individual standard deviations at a 95% confidence interval.

Regarding the mid-term effect, 14 weeks after fertilizing, initial (week 7) differences in soil faunal feeding activity converged and became more even between treatments. However, the reduction was more evident among those treatments with higher feeding activity during the initial weeks (Figures 1B and 2). Hence, the average feeding activity was 62.61% among all treatments seven weeks after fertilizing, which dropped significantly ($p = 0.001$) to 47.75% at the mid-term evaluation (Figure S1, Table S1).

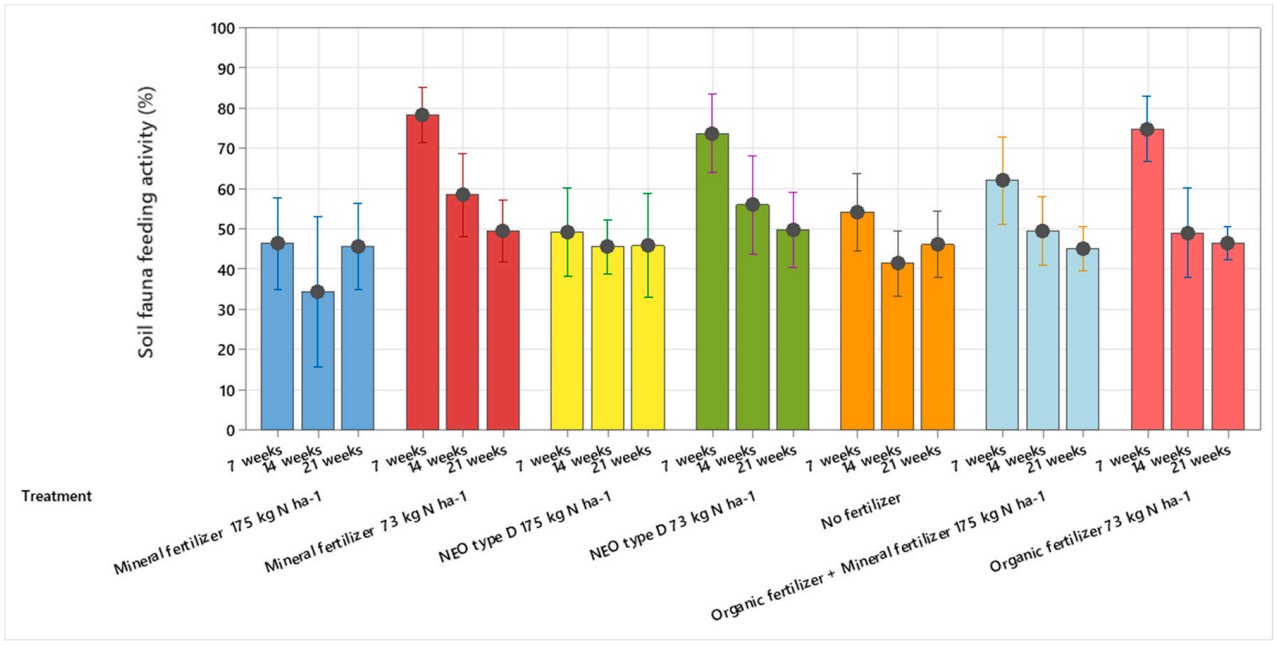

**Figure 2.** Early (0–7 weeks), mid-term (7–14 weeks), and late effects (14–21 weeks) on soil fauna feeding activity (%) between different fertilization treatments. Error bars are individual standard deviations at a 95% confidence interval.

Furthermore, at mid-term evaluation, feeding activities were not significantly different ($p = 0.08$) among fertilization treatments. Instead, mineral fertilizer 73 kg N ha$^{-1}$ (58.44%), NEO type D 73 kg N ha$^{-1}$ (55.99%), organic + mineral fertilizer 175 kg N ha$^{-1}$ (49.48%), organic fertilizer 73 kg N ha$^{-1}$ (48.96%), and NEO type D 175 kg N ha$^{-1}$ (45.57%) had higher soil faunal feeding activity than no fertilizer (41.41%). By comparison, mineral fertilizer 175 kg N ha$^{-1}$ (34.38%) had the lowest feeding activity. Thus, only the mineral fertilizer with a high N content reduced the ability of soil fauna to feed; however, as mentioned earlier, this was not statistically significant (Figure 1B, Table S1).

The late fertilization effect on feeding activity resembled the mid-term effect, i.e., despite a slight insignificant average reduction from mid-term to late effect among all fertilizing treatments (47.75% to 46.88%), soil fauna feeding activity appeared to stabilize seven weeks after fertilization without showing any significant effects (Figures 2 and S1).

The lower amounts of fertilizer, regardless of fertilizer type, supported higher feeding activity, NEO type D 73 kg N ha$^{-1}$ (49.74%), mineral fertilizer 73 kg N ha$^{-1}$ (49.48%), organic fertilizer 73 kg N ha$^{-1}$ (46.35%) showed higher feeding activity than no fertilizer (46.09%). On the other hand, higher fertilizer amounts, NEO type D 175 kg N ha$^{-1}$ (45.83%), mineral fertilizer 175 kg N ha$^{-1}$ (45.57%), and organic + mineral fertilizer 175 kg N ha$^{-1}$ (45.05%) had lower feeding activity. However, the difference between the highest and lowest feeding activities was a maximum of 4.2% and insignificant (Figure 1C, Table S1).

Lastly, low amounts of NEO type D, mineral fertilizer, organic fertilizer, and to some extent, the combination of organic and mineral fertilizer seemed to stimulate soil fauna feeding activity in the initial weeks after fertilization. However, this early effect gradually disappeared, whereas other treatments, including no fertilizer, had more or less constant soil faunal feeding activity throughout the experiment (Figure 2).

### 3.2. The Abundance of Springtails (Collembola)

We investigated and compared the effects of different fertilization treatments on the abundance of springtails at two field locations; one under cereal and another under grass cultivation. Both fields were fertilized for two consecutive years. Moreover, two samplings were performed, once just before fertilization in early summer and another during fall.

During summer, the abundance of springtails in the cereal field was slightly higher for organic fertilizer than NEO and mineral fertilizer; 509.4, 467, and 297.1 per m$^2$, respectively (Figure 3A, Table S1). However, the difference was insignificant ($p = 0.25$). The same pattern was observed during the fall. The number of springtails was slightly higher for organic fertilizer and no fertilizer than NEO and mineral fertilizer; 213.3, 213.3, 169.8, and 169.8 per m$^2$, respectively (Figure 3B, Table S1). Likewise, the difference between fertilization treatments in the fall sampling was insignificant ($p = 0.669$).

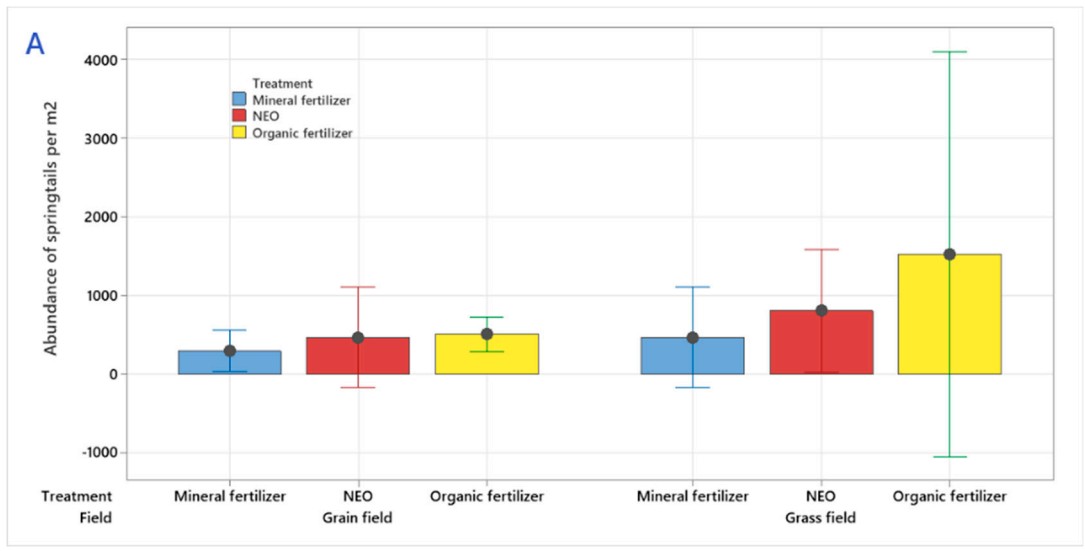

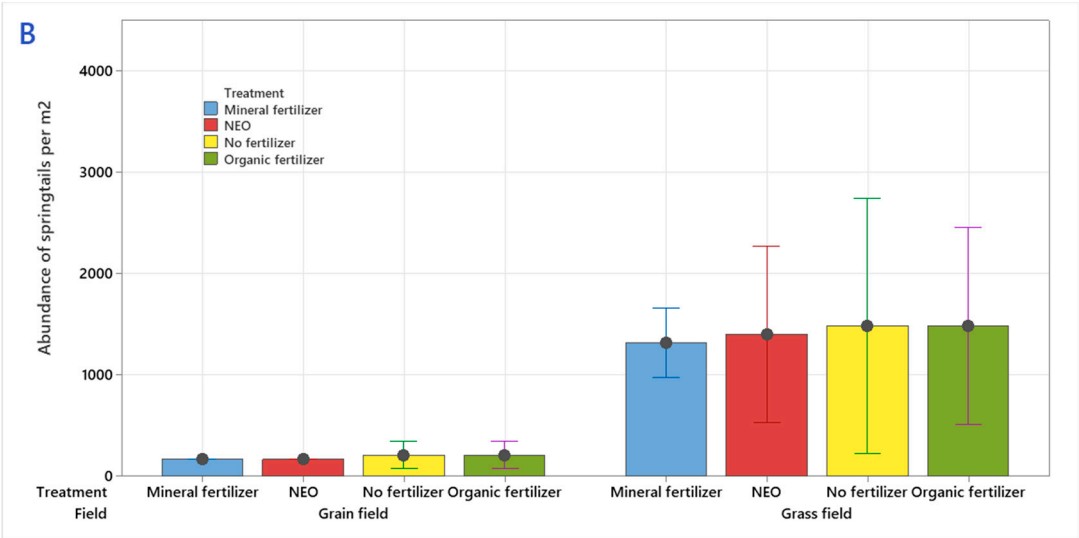

**Figure 3.** The effect of different fertilization treatments on the abundance of springtails per m$^2$ in (**A**) summer and (**B**) fall samplings. Error bars are individual standard deviations at a 95% confidence interval.

Similarly, no fertilization effects were found regarding springtail abundance in the grass field during summer and fall ($p = 0.404$, $p = 0.943$). However, during summer, higher abundance was observed for organic fertilizer than NEO and mineral fertilizer; 1528, 807, and 467 per m$^2$, respectively (Figure 3A, Table S1). In the fall, a slightly higher abundance

was observed for organic fertilizer and no fertilizer than NEO and mineral fertilizer; 1486, 1486, 1401, and 1316, respectively (Figure 3B, Table S1).

Generally speaking, the springtail abundance was higher in the grass field during fall than in summer and almost identical in the grain field during both seasons. Nevertheless, none of the fertilizer treatments affect springtail abundance regardless of field and season.

### 3.3. The Fate of Earthworms (Lumbricidae)

In the 2021 trial, the abundance of earthworms for all treatments increased after eight days, except for the treatment with no fertilizer. However, the difference was insignificant among fertilizing treatments ($p$ = 0.38). Organic fertilizer showed an average increase of 4.33 worms, followed by mineral fertilizer, 4, and NEO, 2.33, while no fertilizer had one fewer living earthworm than the beginning (Figure 4A, Table S1). Moreover, like for abundance, a similar pattern was observed for the average weight change. Among all captured earthworms, mineral fertilizer had an increment of 4.07 g after eight days, followed by organic fertilizer 2.87 g, NEO 1.93 g; however, the no fertilizer control had 1.67 g fewer earthworms (Figure 4A, Table S1). Nevertheless, the difference in weight change among fertilizing treatments was insignificant ($p$ = 0.34).

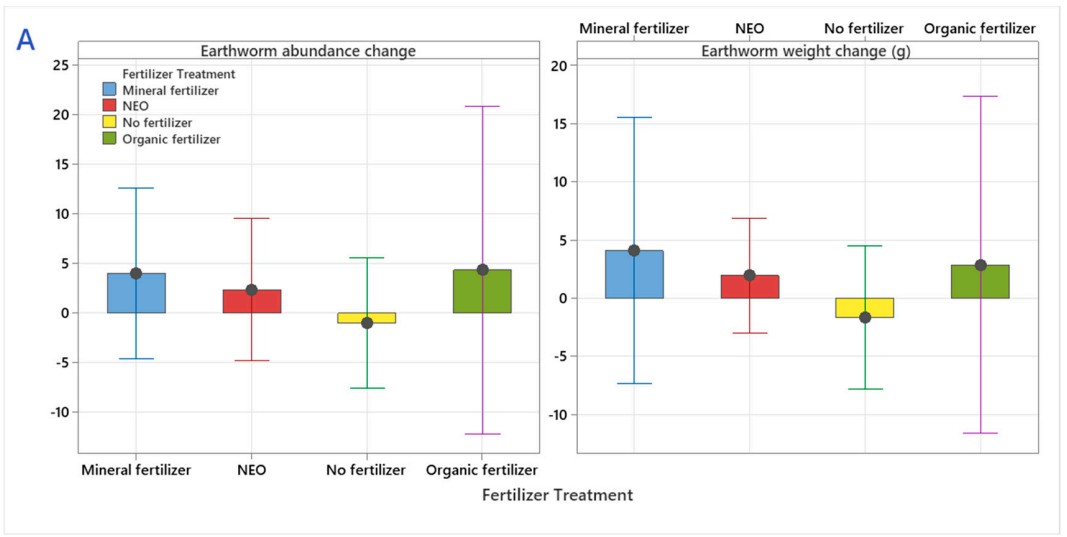

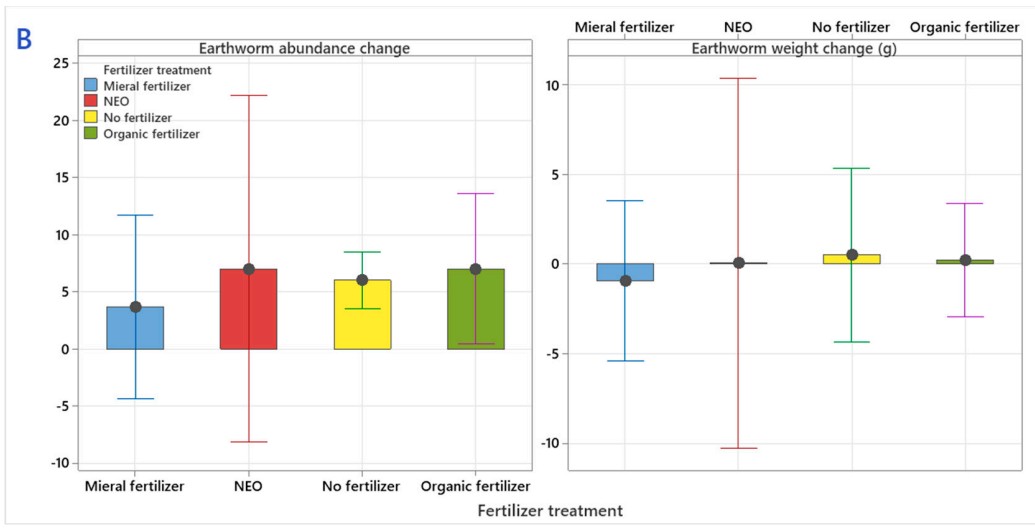

**Figure 4.** The immediate effect of different fertilization treatments on the abundance (left) and weight (right) of earthworms in (**A**) June 2021 and (**B**) June 2022. Error bars are individual standard deviations at a 95% confidence interval.

In the 2022 trial, however, the outcomes were slightly different. Although the abundance was increased for all fertilization treatments, the weights of living worms were almost identical to the beginning. The average abundance increment for organic fertilizer and NEO was 7, no fertilizer 6, and mineral fertilizer 3.67 earthworms (Figure 4B, Table S1). However, the average weight of all earthworms increased by 0.5 g for no fertilizer, 0.22 g for organic fertilizer, and 0.05 g for NEO. In comparison, mineral fertilizer reduced the total weight by 0.09 g (Figure 4B, Table S1). Nevertheless, changes in abundance or weight were insignificant between fertilizing treatments ($p = 0.69$ and $p = 0.83$, respectively).

Thus, the results indicated no adverse effects of fertilizing on the abundance and weight of earthworms, regardless of the fertilizer type used in both experiments.

## 4. Discussion

It is known that N fertilization affects the taxonomic composition of soil faunal communities, their population dynamics, and their feeding activity. However, it is not well understood if soil-dwelling organisms adapt to these external factors [59], especially when the external factor is a newly developed fertilizer (NEO). In this study, we investigated and compared the effects of different fertilization regimes on soil-dwelling organisms. We screened changes in soil faunal feeding activity under controlled conditions, the abundance of springtails, and the immediate effect of different fertilizers on the abundance and weight of earthworms under field conditions. The goal was to detect if NEO, a novel fertilizer with potentially toxic contents of nitrite, has any detrimental effects on soil-dwelling organisms compared to conventional fertilizers.

### 4.1. Soil Fauna Feeding Activity

The soil fauna feeding activity was evaluated at three intervals in pot experiments under controlled conditions; 0–7 weeks (early effect), 7–14 weeks (mid-term effect), and 14–21 weeks (late effect). Low (73 kg N ha$^{-1}$) and high (175 kg N ha$^{-1}$) rates of mineral fertilizer, NEO, organic fertilizer (untreated slurry), and a mixture of organic and mineral fertilizer were used as the fertilizing treatments.

Early effect analysis showed that low doses of fertilizer stimulated feeding activity irrespective of fertilizer type, while high amounts of fertilizer resulted in slightly less feeding activity. The only exception was the combination of organic and mineral fertilizer, which tended to have a higher feeding activity than no fertilizer. In line with our results, a grassland study showed that a high amount of organic fertilizer reduced soil fauna feeding activity within days after fertilizing [59]. Except for this, we could not detect any beneficial or detrimental early effect of fertilizers on soil fauna feeding activity. It may be argued that microbial biomass is promoted within the first weeks after high N fertilization resulting in alternate food sources for soil mesofauna, and they may have shifted away from the bait substrate [59], which explains lower feeding activity under higher fertilization.

The mid-term evaluation showed almost the same pattern as the evaluation of the early effect. The only difference was that at the higher N application rates, given organic and mineral fertilizer combination showed a slightly higher feeding activity than the organic fertilizer alone. Additionally, compared to no fertilizer, a high concentration of NEO showed a slightly increased soil faunal feeding activity. However, variations across fertilization treatments were smaller at the mid-term assessment than at the early effect evaluation, demonstrating that the initial stimulation gradually faded with time after fertilizing.

Finally, at the late effect evaluation, the initial stimulation by low amounts of fertilizer disappeared. Like during the early effect evaluation, higher amounts of fertilizer had lower feeding activities, and lower amounts of fertilizer had higher feeding activity irrespective of fertilizer type; whereas the difference among treatments was much smaller than in short- and mid-term evaluations, with less than five percent difference between the highest and lowest feeding activities.

Although similar to an earlier study [34], higher amounts of fertilizer, regardless of fertilizer type, initially showed a somewhat negative effect on soil faunal feeding activity,

this detrimental effect progressively stabilized with time after fertilization. Furthermore, after some weeks of fluctuations in soil faunal feeding activity, a similar stabilizing effect has been reported in an oil palm plantation fertilized with different amounts of mineral N fertilizer [26]. The rationale for these transient effects might be the soil's buffering capacity and other soil chemical responses that gradually diminish fertilization's perturbation effect. Moreover, the soil fauna may be functionally redundant, conveying resilience to transient perturbations. Thus, we can summarize that neither NEO nor conventional fertilizers used in our experiment adversely affected the soil faunal feeding activity.

## 4.2. The Abundance of Springtails (Collembola)

In the grain field during summer, some weeks after fertilization, the numbers of springtails were almost identical among the plots fertilized with organic fertilizer and NEO. There were slightly fewer springtails than the latter two in the plots fertilized with mineral fertilizer. However, in the grass field, the plots fertilized with organic fertilizer supported an almost double and quadruple number of springtails compared to the plots fertilized with NEO and mineral fertilizer, respectively. Correspondingly, in the cereal field during fall, there were slightly more springtails in the plots fertilized with organic fertilizer and no fertilizer than in the plots fertilized with NEO and mineral fertilizer. The same pattern was observed in the grass field during the fall.

NEO had no adverse effects on the number of springtails after fertilization or after harvest; the number of springtails was generally lower during the summer than during the fall. It has been indicated that the abundance of springtails decreases after cattle slurry application [32]. However, this does not have been the case in our study. There might be two reasons for the apparent lack of response to organic fertilization. Springtails are moisture-dependent organisms [37]; therefore, it is a valid argument that sampling on a warm sunny Scandinavian day with limited moisture in the surface soil forced a major part of the springtail community to move deeper in the soil to avoid desiccation [60]. Moreover, more decaying plant matter as food for springtails may be available deeper in the soil after harvest [61].

Nonetheless, in line with our findings, another study showed almost no fertilization effect on the abundance of springtails [62], while another study indicated that fertilization increases the abundance of springtails [63]. Our study showed no adverse effect of NEO or other fertilizers on the abundance of springtails.

## 4.3. The Fate of Earthworms (Lumbricidae)

We used our developed method to investigate and compare the immediate effects of NEO and other fertilizers on earthworms. The experiment was repeated twice, once during summer 2021 and then in summer 2022. We targeted both the changes in the abundance and the average total weight of earthworms. The treatments were mineral fertilizer, NEO, organic fertilizer, and no fertilizer.

The results from the first experiment indicated no negative effect of fertilizer treatments on the abundance or weight of earthworms after eight days. Only the treatment with no fertilizer showed a slight reduction in the abundance and weight of earthworms. Moreover, roughly the same results were drawn from the second-year experiment. The only difference was that no fertilizer treatment did not lead to any reduction in the abundance or weight of earthworms.

In the case of NEO and organic fertilizer, a promoting effect of adding organic matter to the soil on the earthworm population was expected [18,30]. However, the concern was that excessive liquid slurry in a single dose might adversely affect earthworms [20]. Nonetheless, this did not occur with the amounts applied in our experiments. Moreover, mineral fertilizers might benefit earthworms through direct or indirect effects [19,20,22,64]. However, ammonia-based fertilizer potentially could have adverse effects on the earthworm population in the long run by lowering soil pH [18,65]; this was not the case in our experiments.

The concern might arise from the increasing number of earthworms in our experimental plots after eight days. Earthworms might have escaped their confinement even though we tested this before the study. Another possibility might be that tiny juveniles were contained in the soil before starting the experiment, which grew larger and became discoverable after eight days. Moreover, the most unlikely scenario might be that there were juveniles hatched from the cocoons within the experimental period. Notwithstanding, it is reasonable to argue that these error sources should have been identical for all experimental plots. Thus, it is logical to conclude that fertilization with NEO or any other fertilizer did not inhibit earthworms in the soil but supported an increase in number and activity.

## 5. Conclusions

NEO, the novel, plasma-treated nitrogen-enriched organic fertilizer, did not adversely affect soil faunal feeding activity, the abundance of springtails, and the abundance and weight of earthworms, as observed in pot and field trials. Moreover, fertilization with organic and mineral fertilizers was not seemed to harm the selected soil-dwelling organisms. Hence, NEO does not adversely affect the selected soil-dwelling organisms compared to conventional fertilization regimes commonly used in plant production today.

**Supplementary Materials:** The following supporting information can be downloaded at: https://www.mdpi.com/article/10.3390/agronomy12102314/s1; Table S1: Effects of different fertilizing treatments, including mineral fertilizer, NEO, organic fertilizer (untreated cattle slurry), organic fertilizer + mineral fertilizer (MF), and no fertilizer on soil fauna feeding activity (%) in the early effect, mid-term, and late effect evaluations, springtail abundance in summer and fall samplings at crop and grass fields, and the abundance and weight change (g) of earthworms, respectively. The Games–Howell pairwise comparison method at a 95% confidence interval is used to compare the differences between means. Means that do not share a letter are significantly different.; Figure S1: Effects of all fertilizing treatments on soil fauna feeding activity (%) at seven weeks, 14 weeks, and 21 weeks after fertilizing. Individual standard deviations at a 95% confidence interval are used in the graphs.

**Author Contributions:** Conceptualization, H.M., T.C. and S.Ø.S.; methodology, T.C., H.M., S.Ø.S. and R.P.; software, H.M.; validation, H.M., T.C., S.Ø.S. and R.P.; formal analysis, H.M.; investigation, H.M. and R.P.; resources, T.C. and S.Ø.S.; data curation, H.M.; writing—original draft preparation, H.M.; writing—review and editing, H.M., S.Ø.S., P.D., T.C. and R.P.; visualization, H.M.; supervision, S.Ø.S. and T.C.; project administration, T.C.; funding acquisition, T.C. and S.Ø.S. All authors have read and agreed to the published version of the manuscript.

**Funding:** This research was funded by the Research Council of Norway, grant number 309640 Plasmabehandlet husdyrgjødsel—gjødselvirkning, miljøpåvirkning og klimagassutslipp, and the APC was funded by Inland Norway University of Applied Sciences.

**Institutional Review Board Statement:** Not Applicable.

**Informed Consent Statement:** Not Applicable.

**Data Availability Statement:** Data is accessible at https://osf.io/n4dyc/?view_only=25142c8ddd854758b8808d14fc792a76 (accessed on 3 August 2022).

**Acknowledgments:** The authors would like to acknowledge the administrative efforts of Morten Tofastrud and Elisabeth Røe throughout the project. Besides, we would like to thank the funders of the project, the Research Council of Norway and N2 Applied, along with those who provided us equipment and infrastructure required for our study: Norwegian Agricultural Extension service (NLR), Inland Norway University of Applied Sciences (INN), and Norwegian Centre for Organic Agriculture (NORSØK).

**Conflicts of Interest:** The authors declare no conflict of interest. The funders had no role in the design of the study; in the collection, analyses, or interpretation of data; in the writing of the manuscript, or in the decision to publish the result.

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
