# Peer review of "Plasma-Treated Nitrogen-Enriched Manure Does Not Impose Adverse Effects on Soil Fauna Feeding Activity or Springtails and Earthworms Abundance"

_agronomy, doi:10.3390/agronomy12102314_

Round 1

Reviewer 1 Report

It was a well-written paper and can be accepted after minor corrections. Here are some comments to improve it.

1. Improve your overall English language by a native.

2. Make the graphs clear and easy to understand even in black and white.

3. Use some recent references.

Reviewer 2 Report

Interesting work and new suject. However i wonder if you can explain me three things:

1/ Soil faune feeding activity: how did you choose the time of sampling (7 weeks, 14 weeks and 21 weeks) ?

2/ how can you explain that in many graphes, treatment "No fertiliser" has at least the same effect than the treatment with fertiliser and some times more?

3/ how can you explain the high standard deviation in all your results? Its pity because it is difficult to discuss the results with such deviation.

I think you should develop better your conclusion.

Reviewer 3 Report

Ligne 175: "Bait-lamina strips": Could you explain it briefly?

ligne 330: how the authors could explain the variation in percent feeding activity of soil fauna during the three time periods (early, mid, and late)?

The authors should improve the discussion and conclusion section.

Could you explain further the novelty of your study?

Could NEO replace organic fertilizer ?

Reviewer 4 Report

In this paper, the author evaluated the effects of NEO, a new fertilizer, on soil organisms and concluded that NEO did not affect soil organisms. This study is innovative, and useful for the agriculture production but it also has some shortcomings.

1. Three experiments were designed in this paper and the first two experiments, including 2.1 and 2.2, has a certain reliability. However, for 2.3, the variance of experimental results is too large, resulting in insignificant differences among treatments, which is insufficient to explain the relationship among these treatments.

2. Although the author has told readers in the materials and methods that two statistical methods were used for difference analysis, it is necessary to mark which statistical method was used for difference analysis and which method was used for multiple comparison under the specific pictures in the paper.

3. Line 134 showed an obvious error. The author said that which in total is 4439 mg L−1 –N.  It's a compound of nitrogen, not N.

Reviewer 5 Report

In material and methods, the treatment of NEO with low voltage plasma should be explained in some detail, which seems to me to be what this type of ionizing plasma apparatus entails, and its effects on manure.  

2.1 Soil fauna feeding activity

It is assumed that the feeding activity of the soil fauna belongs to the fauna existing in the soil of the adjoining farm, of which neither the Collembola nor the lumbricid numbers are known, they could be Oligochaeta or other soil animals. I think it is a gap that at least needs to be explained.

253-263 it seems to me that at a depth of 5 cm by the modified Berlese-Tullgren procedure, not all the collembola will come out, the deeper ones will dry up before reaching the surface below unless the extraction method is modified to keep the humidity in the upper chamber.

In my opinion, the type of collembola, at least at the family level, that is recovering from the soil would be desirable in the supplementary material, since not all of them are related to soil conditions. The springtails of the herbaceous surface part, which are obtained with the 5cm x 5cm core, are not affected by soil conditions, they are epigeal Collembola that should be discounted from the evaluations.

Round 2

Reviewer 3 Report

I think the article might be appropriate for publication.